# A Pan-Canadian Consensus Statement on First-Line PARP Inhibitor Maintenance for Advanced, High-Grade Serous and Endometrioid Tubal, Ovarian, and Primary Peritoneal Cancers

**Anna V. Tinker** [1,2], **Alon D. Altman** [3,4], **Marcus Q. Bernardini** [5,6], **Prafull Ghatage** [7,8], **Lilian T. Gien** [6,9], **Diane Provencher** [10,11], **Shannon Salvador** [12], **Sarah Doucette** [13] and **Amit M. Oza** [6,14,*]

1 Division of Medical Oncology, BC Cancer, Vancouver, BC V5Z 4E6, Canada; ATinker@bccancer.bc.ca
2 Department of Medicine, University of British Columbia, Vancouver, BC V6T 1Z4, Canada
3 Division of Gynecologic Oncology, Cancer Care Manitoba, Winnipeg, MB R3E 0V9, Canada; alon.altman@cancercare.mb.ca
4 Department of Obstetrics Gynecology and Reproductive Sciences, University of Manitoba, Winnipeg, MB R3A 1R9, Canada
5 Division of Gynecologic Oncology, Princess Margaret Cancer Center, University Health Network, Sinai Health System, Toronto, ON M5B 2M9, Canada; Marcus.Bernardini@uhn.ca
6 Department of Obstetrics and Gynecology, University of Toronto, Toronto, ON M5G 1X8, Canada; Lilian.Gien@sunnybrook.ca
7 Department of Gynecologic Oncology, Tom Baker Cancer Centre, Calgary, AB T2N 4N2, Canada; prafull.ghatage@albertahealthservices.ca
8 Department of Gynecological Oncology, University of Calgary, Calgary, AB T2N 1N4, Canada
9 Division of Gynecologic Oncology, Odette Cancer Centre, Sunnybrook Health Sciences Centre, Toronto, ON M4N 3M5, Canada
10 Institut du cancer de Montréal, Centre Hospitalier de l'Université de Montréal (CHUM), Montréal, QC H2X 0A9, Canada; diane.provencher.med@ssss.gouv.qc.ca
11 Division of Gynecologic Oncology, Université de Montréal, Montréal, QC H3T 1J4, Canada
12 Department of Obstetrics and Gynecology, McGill University Jewish General Hospital, Montreal, QC H3T 1E2, Canada; shannon.salvador@mcgill.ca
13 IMPACT Medicom Inc., Toronto, ON M6S 3K2, Canada; sarah@impactmedicom.com
14 Division of Medical Oncology and Hematology, Princess Margaret Cancer Centre, University Health Network, Toronto, ON M5G 2C1, Canada
* Correspondence: Amit.Oza@uhn.ca

**Abstract:** The majority of patients with advanced, high-grade epithelial-tubo ovarian cancer (EOC) respond well to initial treatment with platinum-based chemotherapy; however, up to 80% of patients will experience a recurrence. Poly(ADP-ribose) Polymerase (PARP) inhibitors have been established as a standard of care maintenance therapy to prolong remission and prevent relapse following a response to first-line platinum-chemotherapy. Olaparib and niraparib are the PARP inhibitors currently approved for use in the first-line maintenance setting in Canada. Selection of maintenance therapy requires consideration of patient and tumour factors, presence of germline and somatic mutations, expected drug toxicity profile, and treatment access. This paper discusses the current clinical evidence for first-line PARP inhibitor maintenance therapy in patients with advanced, high-grade EOC and presents consensus statements and a treatment algorithm to aid Canadian oncologists on the selection and use of PARP inhibitors within the Canadian EOC treatment landscape.

**Keywords:** epithelial ovarian cancer; PARP inhibitors; BRCA; homologous recombination repair

## 1. Introduction

Although ovarian cancer is the eighth most diagnosed cancer among Canadian women, with 3100 new cases estimated to be diagnosed in 2020, it is the fifth leading cause of cancer-related death in this population [1]. The generally poor prognosis for ovarian

cancer is a result of its early and rapid intraabdominal spread with non-specific symptom presentation that causes most patients to be diagnosed with advanced disease [2]. For epithelial tubo-ovarian cancers (EOC), which account for approximately 90% of ovarian cancer cases, 5-year relative survival rates drop from 93% for those diagnosed with localized disease to 31% for those with distant metastases [3]. Furthermore, tubo-ovarian high-grade serous carcinomas (HGSC), which represent over 75% of EOCs, primarily present with advanced-stage disease and are associated with the poorest prognosis among EOC sub-types [4].

Cytoreductive surgery (primary or interval) and platinum-based chemotherapy have been the mainstay treatment for EOC, fallopian tube, and primary peritoneal cancer (as these are biologically alike, they are referred to synonymously as EOC throughout the manuscript). High-grade EOCs, which include serious or endometrioid cancers, are particularly susceptible to the cytotoxic effects of platinum-based chemotherapy [5,6]. This may in part be due to defects in the homologous recombination DNA repair (HRR) pathway that occur in approximately 50% of high-grade EOCs through different mechanisms, including pathogenic mutations in the BRCA1 and BRCA2 genes. In the context of HRR deficiency (HRD), the DNA crosslinks caused by platinum-chemotherapy cannot be repaired and thus trigger cell death pathways [7]. Despite excellent response to front-line treatment, up to 80% of patients with advanced EOC will experience a recurrence [2], highlighting the need for additional interventions that can maintain remissions and prevent relapse.

Targeted therapies against poly(ADP-ribose) polymerase (PARP) have been established as a standard of care maintenance treatment for patients with advanced EOC who display a response to first-line platinum-chemotherapy. Data demonstrate improved progression-free survival (PFS) of oral maintenance PARP inhibitors over placebo control in this patient population, with tolerable side effects and acceptable safety profiles [8,9]. The subgroup of patients with either germline (inherited) or somatic (tumour acquired) BRCA1/2 mutations derive the greatest benefit from PARP inhibitors. This has contributed to the establishment of BRCA1/2 mutation testing for germline and somatic mutations as standard of care for high-grade EOC in Canada [10]. PARP inhibitors trap PARP proteins at the site of single-strand DNA breaks, disrupting the single-strand break repair pathway and leading to double-strand DNA breaks. When the tumour harbours a BRCA1/2 mutation (germline or somatic), the resulting HRD prevents the repair of double-stranded DNA breaks, which accumulate to the point of genomic catastrophe and trigger cancer cell death [11]. In 2019, olaparib was the first PARP inhibitor to be approved by Health Canada in the first-line maintenance setting as monotherapy for patients with advanced, high-grade EOC who harbour mutations in BRCA1/2 genes and have had a partial or complete response to first-line platinum-based chemotherapy [12]. HRD often occurs in high-grade EOC with wild type BRCA1/2. Accordingly, PARP inhibitors also benefit molecularly unselected patients after first-line therapy as long as there is evidence of responsiveness to first-line platinum-based therapy. In 2020, the PARP inhibitor niraparib was approved by Health Canada for the monotherapy maintenance treatment of patients with advanced, high-grade EOC who are in a complete or partial response to first-line platinum-based chemotherapy, irrespective of their BRCA status [13].

With two different PARP inhibitors on the market in Canada for first-line maintenance treatment of advanced EOC, physicians must choose which is best for their individual patients. As there are no trials directly comparing olaparib and niraparib, PARP inhibitor selection requires some consideration of patient and tumour factors, as well as the expected drug toxicity. This pan-Canadian panel of ovarian cancer experts has developed the following consensus statements and treatment algorithm to aid Canadian oncologists on the selection and use of PARP inhibitors as maintenance therapy in the first-line treatment of advanced, high-grade EOC, which reflect both current clinical evidence and the Canadian treatment landscape.

## 2. Consensus Process

A panel of gynecologic oncologists and medical oncologists specializing in the treatment of gynecologic cancers from across Canada discussed standards of care related to PARP inhibitor maintenance therapy in the first-line setting for patients with advanced, high-grade EOC, as well as current gaps in knowledge and access to testing and treatments. A list of statements was proposed by the lead and corresponding authors, which covered the value of genetic testing to inform PARP inhibitor strategies, considerations for selecting a PARP inhibitor, and duration of PARP inhibitor treatment. Through virtual discussions, the statements were further modified, and consensus was reached. The consensus statements presented reflect evidence from currently published studies and the combined experience of the authors with PARP inhibitor therapy (Table 1).

**Table 1.** Summary of consensus statements for the selection and use of Poly(ADP-ribose) polymerase (PARP) inhibitor maintenance therapy after first-line therapy in patients with advanced, high-grade epithelial tubo-ovarian cancer (EOC).

---

**Genetic Testing to Inform PARP Inhibitor Maintenance Strategies**

(a) All patients with high-grade EOC should have BRCA1/2 mutation testing to:
  i.   Inform hereditary cancer predisposition and the need for cascade testing of family members;
  ii.  Guide first-line PARP inhibitor maintenance in advanced stage cases.

(b) Tumour HRD status is a predictive biomarker of treatment benefit from PARP inhibitors, and testing should be publicly funded.

(c) Assessment of mutations in HRR genes other than BRCA1/2 should not be used as a substitute for HRD testing.

**Selection of PARP inhibitors as first-line maintenance therapy in advanced EOC**

(d) All BRCA1/2-mutated patients with advanced EOC should receive maintenance therapy with a PARP inhibitor following a response to platinum-based chemotherapy. The choice of PARP inhibitor is influenced by several factors, including the expected toxicity profile of each agent.

(e) Patients with advanced EOC who are BRCA1/2 wild-type and have responded to platinum-based chemotherapy should be considered for maintenance treatment with niraparib.

(f) There is evidence to support the combination of olaparib with bevacizumab as a maintenance regimen in patients with advanced, high-grade, HRD-positive EOC who respond to first-line treatment with platinum chemotherapy and bevacizumab.

**Dosing and duration of PARPi maintenance therapy**

(g) Olaparib should be given orally at a starting dose of 300 mg, twice-daily for up to two years in patients with a response to first-line platinum-based chemotherapy. Treatment beyond 2 years should only be considered in patients who have evidence of disease at the 2-year time point for whom ongoing treatment is felt to be beneficial.

(h) Niraparib should be given orally at a starting dose of 200 mg, once-daily for patients weighing less than 77 kg or with a platelet count of less than 150,000/μL, or at a starting dose of 300 mg, once-daily for patients weighing greater than or equal to 77 kg or with a platelet count of greater than or equal to 150,000/μL, for up to three years. Treatment beyond 3 years should only be considered in patients who have evidence of disease at the 3-year time point for whom ongoing treatment is felt to be beneficial.

(i) Patients should be informed of the expected treatment duration and data to support completion of treatment at the time of maintenance therapy initiation.

(j) Routine clinical assessments and laboratory monitoring are required, for the duration of therapy, taking into consideration the common adverse events.

(k) Toxicities can be managed through dose interruptions and reductions as described in the product monograph for each PARP inhibitor, followed by a rechallenge upon resolution of toxicity.

(l)   Switching between approved PARP inhibitors in the first-line maintenance setting for unmanageable toxicity is considered a reasonable option to allow for the continuation of PARP inhibitor therapy.

EOC, epithelial tubo-ovarian cancer; HRD, homologous recombination repair deficiency; HRR, homologous recombination repair; PARP, poly(ADP-ribose) polymerase.

## 3. Genetic Testing to Inform PARP Inhibitor Maintenance Strategies

*3.1. Consensus Statements*

(a)   All patients with high-grade EOC should have BRCA1/2 mutation testing to:

  i.    Inform hereditary cancer predisposition and the need for cascade testing of family members;
  ii.   Guide first-line PARP inhibitor maintenance in advanced stage cases.

(b)   Tumour HRD status is a predictive biomarker of treatment benefit from PARP inhibitors, and testing should be publicly funded.

(c)   Assessment of mutations in HRR genes other than BRCA1/2 should not be used as a substitute for HRD testing.

*3.2. Summary of Evidence*

Germline (inherited) mutations in BRCA1 or BRCA2 genes are found in 5–15% of patients with EOC and in up to 15–25% of patients with HGSC depending on the population studied [14–19]. Tumour acquired (somatic) mutations in BRCA1/2 genes have also been reported to occur in approximately 6–7% of cases [15,20,21]. Patients with EOC and BRCA1/2 mutations have better 5-year overall survival rates than those who are BRCA1/2 wild-type, with the best prognosis reported for patients with BRCA2 alterations [18,22–24]. Pathogenic BRCA1/2 mutations are also strong predictors for benefit from PARP inhibitors, as demonstrated by several clinical trials in both the first-line [8,9] and recurrent disease settings [25–29] (Table 2). This holds true for both germline and somatic BRCA1/2 mutations [30,31].

In addition to the prognostic and predictive value of BRCA1/2 status, the identification of germline BRCA1/2 mutations benefit relatives, allowing for cascade testing and identification of unaffected carriers. Germline BRCA1/2 mutations are associated with increased susceptibility to EOC and breast cancer [32], with a lifetime risk of developing EOC estimated at 40–60% for BRCA1 and 11–27% for BRCA2 mutation carriers, compared with 1.3% of women in the general population [2,33]. BRCA1/2 carriers can consider surgical prophylactic strategies such as bilateral salpingo-oophorectomy and mastectomy, as well as enhanced breast cancer screening [32,34,35]; strategies demonstrated to reduce the risk of cancer related death in these high-risk individuals.

Although BRCA1/2 mutations are enriched in patients with EOC who are <50 years old at diagnosis and/or have a family history of ovarian or breast cancer, restricted testing criteria will miss a significant portion of BRCA1/2 germline mutation carriers [15]. As such, the American Society of Clinical Oncology (ASCO), the European Society for Medical Oncology (ESMO) in conjunction with the European Society of Gynaecological Oncology, and the Society of Gynecologic Oncology of Canada (GOC) recommend that all patients with high-grade EOC undergo testing for BRCA1 and BRCA2 mutations [10,34,36].

Germline BRCA1/2 testing can be additionally advantageous if performed within a multigene panel that assesses other genes linked to cancer predisposition. Genes involved in DNA repair pathways which have been confirmed or are being investigated as EOC susceptibility genes include RAD51C, RAD51D, BRIP1, MLH1, MSH2, MSH6, PMS2, and PALB2, detection of which may provide useful information to patients and their families [34,37].

Apart from BRCA1/2, mutations in other genes involved in the HRR pathway are present in up to 9% of EOCs and are associated with improved overall survival and responsiveness to platinum chemotherapy [21,38]. Small retrospective analyses of trials on

recurrent high-grade EOC have suggested that patients with mutations in other HRR genes may have a PFS benefit from PARP inhibitors [39,40]. However, given the small number of patients evaluated in these studies, there remains uncertainty about the predictive and prognostic value of other HRR genes in EOC [41]. There are no data quantitating the impact of individual HRR gene mutations on response to PARP inhibitor maintenance in first-line EOC. Thus, mutation status of these genes alone is not sufficient to inform PARP inhibitor treatment.

The HRD phenotype is estimated to be present in approximately 50% of EOCs [42] and is associated with a benefit from PARP inhibitor maintenance, independent of BRCA1/2 mutation status. This has been demonstrated in preplanned exploratory analyses of several clinical trials evaluating PARP inhibitor maintenance in the first-line and recurrent settings, with results varying by HRD testing method and maintenance strategy used [9,27,29,43,44] (Table 2). The predictive value of HRD testing is validated only in the settings of first-line therapy with platinum response and platinum-sensitive and platinum-responsive recurrent disease. The utility of this test and measurement of genomic scar may change over time following repeated therapy, thus an HRD score may not be meaningful in the setting of platinum resistant EOC.

The PRIMA study is the only phase III trial of a PARP inhibitor given as switch maintenance following a response to first-line platinum-chemotherapy which reported PFS stratified by HRD status [9]. In this study HRD was determined using the Myriad myChoice® test (Myriad Genetics, Inc., Salt Lake City, UT, USA), which evaluates BRCA1/2 mutation status and calculates a composite genomic instability score (GIS), based on loss of heterozygosity, telomeric allelic imbalance, and large-scale transitions. PRIMA reported a significant PFS improvement for niraparib in both the HRD/BRCA1/2 wild-type and homologous recombination proficiency (HRP) populations, using an HRD GIS cut-off of ≥42. However, the magnitude of PFS benefit was greater for the HRD population compared to the HRP population. Presently, ASCO and ESMO guidelines do not recommend the routine use of HRD testing in EOC on the basis that current assays cannot sufficiently differentiate which patients respond to PARP inhibitors [34,36]. However, National Comprehensive Cancer Network guidelines for ovarian cancer acknowledge that HRD testing for patients with EOC may be considered [45].

### 3.3. Interpretation and Canadian Perspective

The value of BRCA1/2 mutation testing as a prognostic tool, as a predictive biomarker for response to PARP inhibitor therapy, and as a means to trigger familial cascade testing for inherited germline mutations followed by risk-reducing strategies for identified BRCA1/2 carriers is established. Germline assessment alone will miss approximately 7% of patients who harbour somatic BRCA1/2 mutations [20] and tumour testing alone may miss approximately 5% of patients who have germline mutations due to decreased test sensitivity and coverage [46]. The panel recommends testing for both germline and somatic BRCA1/2 mutations in all patients with high-grade EOC. This recommendation is also supported by the GOC [10,47].

The sequencing of germline and somatic BRCA-testing should take into account available resources and timeliness of testing. In some Canadian jurisdictions, wait times for pre-test genetic counselling can be 3–5 months [48], meaning the BRCA mutation status may remain unknown at the completion of first-line treatment. Given the requirement of a confirmed BRCA1/2 mutation for patients to access olaparib in the first-line maintenance setting in Canada, failure to perform timely assessment and reporting of BRCA1/2 mutation status may limit maintenance treatment options. Reflex tumour testing of BRCA1/2 mutations at the time of diagnosis of advanced EOC can allow for a turnaround of 3–4 weeks and can detect the majority of germline mutations and is thus encouraged to optimize testing efficiency and increase the rate of BRCA1/2 mutation detection [47].

The panel recognizes that current testing methods for HRD status have limitations in identifying patients who might benefit from PARP inhibitors. Niraparib maintenance

therapy demonstrated a PFS benefit in the overall population in the PRIMA study and sub-group analyses confirmed the greatest magnitude of benefit was seen in the BRCA1/2 mutated and HRD populations, with a difference in median PFS of less than 3 months for the HRP population. The pan-Canadian Oncology Drug Review's Expert Review Committee has stated that due to the lack of a clinically validated or standardized HRD test, results from PRIMA based on HRD status should be interpreted with caution and that treatment decisions should not be guided based on HRD testing alone [49]. While the panel agrees HRD status should not be the sole factor guiding decisions to use niraparib in the first-line maintenance setting, current commercially available Myriad HRD tests can inform the likely magnitude of benefit of maintenance niraparib. This may allow patients to make informed decisions regarding maintenance treatment, and thus should be accessible to all BRCA1/2 wild-type patients. The cost-effectiveness of therapy in the first-line setting must be considered, particularly in Canada where oncology drugs are publicly funded. Canadian studies comparing the economic impact of providing niraparib treatment only to patients with HRD determined through commercial testing versus all BRCA wild-type advanced EOC patients are needed.

**Table 2.** Progression-free survival results by molecular subgroups from key studies investigating PARP inhibitors as first-line maintenance therapy in advanced EOC.

| Trial Name, Study Phase | Treatment Arms | Study Population | HRD Testing Method | PFS Results (PARP Inhibitor vs. Control) | | |
|---|---|---|---|---|---|---|
| | | | | Population | Median (Months) | HR (95% CI) |
| SOLO-1 [8] (NCT01844986) Phase III | olaparib vs. placebo | Stage III/IV BRCAm EOC following PR/CR to CT | N/A | BRACm(ITT) | 56.0 vs. 13.8 | 0.33 (0.25–0.43) |
| PRIMA [9] (NCT02655016) Phase III | niraparib vs. placebo | Stage III/IV high-risk EOC with visible residual disease following PR/CR to CT | Myriad myChoice CDx (HRD = GIS ≥42 or BRCA1/2 mutation) | ITT | 13.8 vs. 8.2 | 0.62 (0.50–0.76) |
| | | | | HRD | 21.9 vs. 10.4 | 0.43 (0.31–0.59) |
| | | | | BRCAm | 22.1 vs. 10.9 | 0.40 (0.27–0.62) |
| | | | | HRD/BRCAwt | 19.6 vs. 8.2 | 0.50 (0.31–0.83) |
| | | | | HRP | 8.1 vs. 5.4 | 0.68 (0.49–0.94) |
| PAOLA1 * [44] (NCT02477644) Phase III | olaparib + bevacizumab vs. placebo + bevacizumab | Stage III/IV EOC following PR/CR to CT + bevacizumab | Myriad myChoice CDx (HRD = GIS ≥42 or BRCA1/2 mutation) | ITT | 22.1 vs. 16.6 | 0.59 (0.49–0.72) |
| | | | | HRD | 37.2 vs.17.7 | 0.33 (0.25–0.45) |
| | | | | BRCAm | 37.2 vs. 21.7 | 0.31 (0.20–0.47) |
| | | | | HRD/BRCAwt | 28.1 vs. 16.6 | 0.43 (0.28–0.66) |
| | | | | HRP/HRnd | 16.9 vs. 16.0 | 0.92 (0.72–1.17) |
| VELIA *,†,‡ [50] (NCT02470585) Phase III | CT + veliparib → veliparib vs. CT + 66veliparib → placebo vs. CT + placebo → placebo | Stage III/IV high-grade serous ovarian carcinoma | Myriad myChoice CDx (HRD = GIS ≥33 or BRCA1/2 mutation) | ITT | 23.5 vs. 17.3 | 0.68 (0.56–0.83) |
| | | | | HRD | 31.9 vs. 20.5 | 0.57 (0.43–0.76) |
| | | | | BRCAm | 34.7 vs. 22.0 | 0.44 (0.28–0.68) |
| | | | | BRCAwt | 18.2 vs. 15.1 | 0.80 (0.64–1.00) |
| | | | | HRP | 15.0 vs. 11.5 | 0.81 (0.60–1.09) |

CI, confidence interval; CR, complete response; CT; chemotherapy; EOC, epithelial tubo-ovarian cancer; HR, hazard ratio; HRD, homologous recombination repair deficiency; HRP, homologous recombination repair proficiency; ITT, intent-to-treat; m, mutated; N/A not applicable; PFS, progression-free survival; PR, partial response; wt, wild-type. * Investigated treatment not approved by Health Canada. † This study investigated veliparib added to first-line induction with chemotherapy and as maintenance therapy. ‡ Chemotherapy + veliparib treatment followed by veliparib maintenance vs. chemotherapy + placebo treatment followed by placebo maintenance.

## 4. Selection of PARP Inhibitors as First-Line Maintenance Therapy in Advanced EOC

*4.1. Consensus Statements*

(d)   All BRCA1/2-mutated patients with advanced EOC should receive maintenance therapy with a PARP inhibitor following a response to platinum-based chemotherapy. The choice of PARP inhibitor is influenced by several factors, including the expected toxicity profile of each agent.

(e)   Patients with advanced EOC who are BRCA1/2 wild-type and have responded to platinum-based chemotherapy should be considered for maintenance treatment with niraparib.

(f)   There is evidence to support the combination of olaparib with bevacizumab as a maintenance regimen in patients with advanced, high-grade, HRD-positive EOC who respond to first-line treatment with platinum chemotherapy and bevacizumab.

*4.2. Summary of Evidence*

Olaparib and niraparib are approved in Canada as monotherapy maintenance treatment of advanced, high-grade EOC after a response to first-line platinum chemotherapy based on the phase III SOLO-1 and PRIMA trials, respectively. These two trials, while both of maintenance therapy, differ in design and study populations [8,9] (Table 3) There is no existing trial comparing olaparib with niraparib. In SOLO-1, 391 patients with newly diagnosed stage III or IV BRCA1/2 mutated EOC were randomized 2:1 to receive either olaparib or placebo twice-daily for two years or longer following a complete or partial response to first-line platinum-chemotherapy [8]. At a median follow-up of five years, the primary endpoint of investigator-assessed PFS was significantly improved for olaparib (hazard ratio [HR] 0.33; 95% confidence interval [CI], 0.25–0.43), with the median PFS reaching 56.0 months for olaparib versus 13.8 months for placebo [8]. A PFS benefit for olaparib over placebo was consistently observed in patients with both high-risk disease (stage IV disease or stage III disease after neoadjuvant chemotherapy and interval cytoreduction or with gross residual disease following initial cytoreductive surgery) and lower-risk disease (optimally debulked stage III disease). Five-year PFS rates in the olaparib and placebo arms were 42.1% versus 17.3% for the high-risk subgroup and 56.0% versus 25.4% for the lower-risk subgroup, respectively [51].

The most common adverse events of any grade reported for olaparib were nausea (78% vs. 38% with placebo) and fatigue/asthenia (64% vs. 42% with placebo) [8]. Anemia (22% vs. 2%) and neutropenia (8% vs. 5%) were the most frequently reported grade ≥3 adverse events with olaparib. These adverse events accounted for the majority of adverse events leading to treatment discontinuation, which occurred in 12% of patients in the olaparib arm and 3% in the placebo arm. No new safety signals were reported at the 5-year follow-up, with rates of myelodysplastic syndromes (MDS) or acute myeloid leukemia (AML) at 1.2% in the olaparib arm (no MDS/AML in placebo arm) [52].

The PRIMA trial enrolled 733 patients with EOC who had stage IV disease or stage III disease that was either inoperable or had visible residual tumour after primary debulking surgery [9]. All patients were randomized 2:1 to receive niraparib or placebo tablets once-daily for 3 years or until disease progression. Two protocol amendments were introduced during the study. The first amendment removed the requirement of HRD to enroll, and the second allowed a personalized starting dose of niraparib based on weight and platelet count. The primary endpoint of PFS assessed by blinded independent central review was performed hierarchically, with the HRD population being tested first, followed by the overall population. At a median follow-up of 13.8 months, a statistically significant improvement in PFS was observed in the niraparib arm compared to the placebo arm in both the HRD population (median PFS: 21.9 months vs. 10.4 months; HR 0.43; 95% CI, 0.31 to 0.59; $p < 0.001$) and overall population (median PFS: 13.8 months vs. 8.2 months; HR 0.62; 95% CI, 0.50 to 0.76; $p < 0.001$). Subgroup analyses within the HRD population demonstrated similar PFS benefit for niraparib over placebo for patients with BRCA1/2

mutations (median PFS: 22.1 vs. 10.9 months; HR 0.40; 95% CI, 0.27 to 0.62) and patients with wild-type BRCA1/2 (median PFS: 19.6 vs. 8.2 months; HR 0.50; 95% CI, 0.31 to 0.83). In addition, a statistically significant but numerically smaller PFS benefit was reported for niraparib versus placebo in the population of patients with HRP (median PFS: 8.1 vs. 5.4 months; HR, 0.68; 95% CI, 0.49–0.94), and in those with a partial response to platinum-based chemotherapy (median PFS: 8.3 vs. 5.6 months; HR, 0.60; 95% CI, 0.43–0.85). A post-hoc analysis stratifying patients by residual disease after primary or interval debulking surgery demonstrated comparable reductions in the risk of progression for niraparib versus placebo across subgroups [53].

Treatment discontinuation due to adverse events occurred in 12.0% of patients in the niraparib arm and 2.5% of patients in the placebo arm and were mostly related to myelo-suppressive events. One patient in the niraparib group developed MDS. The most common adverse events of any grade reported for niraparib were anemia (63% vs. 18% with placebo), nausea (57% vs. 28% with placebo), and thrombocytopenia (46% vs. 4% with placebo). Anemia (31% vs. 2%) and thrombocytopenia (29% vs. < 1%) were also the most frequently reported grade ≥3 adverse events with niraparib.

**Table 3.** Comparison of study design and patient characteristics in phase III trials of olaparib and niraparib maintenance monotherapy in the first-line setting.

| Study Attribute | SOLO-1 (*N* = 391) | PRIMA (*N* = 733) |
|---|---|---|
| Design | International, randomized (2:1), double-blind | |
| Treatment arms | Olaparib vs. placebo | Niraparib vs. placebo |
| Dosing | Olaparib 300 mg twice-daily up to 24 months or until progression for patients in PR | Niraparib 300 mg once-daily * up to 36 months (or until progression for patients in PR) |
| Eligibility criteria | BRCA1/2 mutated No prior bevacizumab Stage III/IV CR/PR to platinum-CT | Stage III inoperable/visible residual disease and stage IV [+] CR/PR to platinum-CT |
| Stage IV | 17% | 35% |
| PDS/NACT-IDS | 63%/35% | 32%/67% |
| NED or CR after platinum-CT | 74% | 69% |
| BRCA1/2 mutated | 100% | 30% |
| HRD testing | None | Myriad myChoice HRD GIS score ≥42 or BRCA1/2 mutation |
| Primary endpoint | PFS Investigator-assessed | PFS Blinded Independent central review HRD and ITT (Hierarchical testing) |

CR, complete response; CT, chemotherapy; GIS, genomic instability score; HRD, homologous recombination repair deficiency; IDS, interval debulking surgery; ITT, intent-to-treat; NACT, neoadjuvant chemotherapy; NED, no evidence of disease; PDS, primary debulking surgery. * The protocol was amended to include a starting dose of 200 mg once-daily for patients less than 77 kg or with a platelet count of less than 150,000/μL. [+] The original protocol restricted eligibility to patients with HRD; however, was amended to remove biomarker criteria at enrollment and added HRR status as a stratification factor during randomization.

The phase III PAOLA-1 study demonstrated the benefit of olaparib maintenance, in combination with bevacizumab, in the first-line setting for patients with platinum-sensitive advanced, high-grade EOC [44]; however, this regimen has not been reviewed by Health Canada. In this trial, a statistically significant improvement in PFS was observed with maintenance olaparib plus bevacizumab compared with placebo plus bevacizumab after response to first-line platinum–taxane chemotherapy plus bevacizumab (median

PFS: 22.1 months vs. 16.6 months; HR 0.59; 95% CI, 0.49–0.72; *p* < 0.001). Subgroup analyses showed a greater magnitude of benefit for olaparib plus bevacizumab in patients with HRD (both including and excluding BRCA1/2 mutations), with no benefit from the addition of olaparib to maintenance therapy observed in patients with HRP or with unknown HRD (Table 2).

Veliparib and rucaparib are other PARP inhibitors that are being investigated for first-line maintenance treatment in advanced EOC, but do not have Health Canada indications in this setting. Veliparib has been evaluated in the phase III VELIA trial; however, results remain difficult to interpret as this trial assessed veliparib as part of both the induction and maintenance regimen without a control arm to assess whether there was added value to using veliparib in the induction stage [50]. Rucaparib continues to be investigated in the phase III ATHENA trial in combination with nivolumab, and may also add to the body of evidence supporting PARP inhibitor therapy in first-line maintenance once the primary analysis is reported [54].

### 4.3. Interpretation and Canadian Perspective

Patients with advanced high-grade EOC who have confirmed germline or somatic BRCA1/2 mutations have demonstrated the greatest benefit to first-line PARP inhibitor maintenance in both the SOLO-1 and PRIMA trials. Given the significant PFS improvements gained in this population and the lack of other comparatively effective regimens in this setting, all patients with BRCA1/2-mutated advanced EOC who are fit to receive PARP inhibitor maintenance after a response to platinum-based chemotherapy in the first-line setting should be offered this therapy. Although PARP inhibitors are available as maintenance therapy in the second-line setting, they should preferably be given as first-line maintenance, based on the improved PFS [8,28], and the lower risk of MDS/AML associated with first-line versus second-line maintenance [55]. The lower rate of long-term complications such as MDS/AML after first line therapy compared to platinum sensitive recurrent disease likely reflects the shorter duration of maintenance, as well as lower exposure to successive rounds of platinum-based chemotherapy.

In the BRCA1/2 mutated population, selecting which PARP inhibitor to use in the maintenance setting should focus on patient factors and entail a discussion with the patient to review side effects, drug dosing and schedule, and the expected duration of therapy. Although there is overlap in adverse events that commonly occur with olaparib and niraparib treatment, olaparib appears to have slightly higher rates of any grade nausea, diarrhea, and fatigue than niraparib, while niraparib appears to have higher rates of any grade and high-grade hematological adverse events, any grade and high-grade hypertension, and low-grade insomnia than olaparib [56]. PARP inhibitor selection may therefore depend on the expected toxicity profile of each agent, given the context of a patient's anticipated tolerability. The oncologist's familiarity and comfort with monitoring and managing these adverse events may also play a role in treatment selection, as well as patient preference for once- versus twice-daily administration.

Although the risk of progression or death for olaparib over placebo in the BRCA1/2-mutated population may appear slightly lower in the SOLO-1 trial than for niraparib in the PRIMA trial, the trials enrolled different populations and the BRCA1/2 outcomes in the PRIMA trial were based upon a subgroup analysis. The long-term follow-up data on the SOLO-1 trial demonstrate that almost half of the olaparib treated population remained progression-free at 5 years, with a median PFS that was 3.5 years longer than in the placebo group. This suggests that in the BRCA1/2 population, treatment with 24 months of maintenance olaparib may be sufficient to induce long-term remission, and possibly even a cure for some women.

All patients with advanced high-grade EOC who do not harbour a BRCA1/2 mutation and who have a response to first line platinum-based chemotherapy should be considered for maintenance niraparib. If HRD testing is available, patients with HRP disease

may have a lower magnitude of benefit from niraparib maintenance. Other treatment options such as observation, participation in a clinical trial, or maintenance bevacizumab can be considered based on patient and disease factors. In the ICON-7 trial, bevacizumab given concurrently with standard chemotherapy demonstrated a significantly prolonged PFS and overall survival compared to standard chemotherapy alone in a subgroup analysis of patients with EOC who had high-risk disease (median PFS: 16.0 months vs. 10.5 months, HR 0·73; 95% CI, 0·61–0·88; restricted mean survival time 39·3 months [37·0–41·7] vs. 34·5 months [95% CI, 32·0–37·0]; log-rank p-value p = 0·03) [57].

For patients with HRD including those with BRCA1/2 mutations, maintenance therapy with bevacizumab plus olaparib following concurrent chemotherapy and bevacizumab, as per the PAOLA-1 protocol, remains an option, particularly in patients with suboptimally debulked disease. However, bevacizumab use in the first-line setting is not consistent across provinces and the Health Canada approved dose is 7.5 mg/kg every 3 weeks (as per the ICON-7 trial) rather than the 15 mg/kg every 3 weeks used in the PAOLA-1 study. Finally, the PAOLA-1 regimen is not approved by Health Canada or currently reimbursed in any province.

## 5. Dosing and Duration of PARP Inhibitor Maintenance Therapy

### 5.1. Consensus Statements

(g) Olaparib should be given orally at a starting dose of 300 mg, twice-daily for up to two years in patients with a response to first-line platinum-based chemotherapy. Treatment beyond 2 years should only be considered in patients who have evidence of disease at the 2-year time point for whom ongoing treatment is felt to be beneficial.

(h) Niraparib should be given orally at a starting dose of 200 mg, once-daily for patients weighing less than 77 kg or with a platelet count of less than 150,000/μL, or at a starting dose of 300 mg, once-daily for patients weighing greater than or equal to 77 kg and with a platelet count of greater than or equal to 150,000/μL. Treatment beyond 3 years should only be considered in patients who have evidence of disease at the 3-year time point for whom ongoing treatment is felt to be beneficial.

(i) Patients should be informed of the expected treatment duration and data to support completion of treatment at the time of maintenance therapy initiation.

(j) Routine clinical assessments and laboratory monitoring are required, for the duration of therapy, taking into consideration the common adverse events.

(k) Toxicities can be managed through dose interruptions and reductions as described in the product monograph for each PARP inhibitor, followed by a rechallenge upon resolution of toxicity.

(l) Switching between approved PARP inhibitors in the first-line maintenance setting for unmanageable toxicity is considered a reasonable option to allow for the continuation of PARP inhibitor therapy.

### 5.2. Summary of Evidence

The dosing suggestions listed in the product monographs for both olaparib and niraparib follow the dosing used in the SOLO-1 and PRIMA trials, respectively [58,59]. In the SOLO-1 trial, the dose of 300 mg twice-daily was relatively well tolerated, with 52% of patients requiring dose interruption and 29% requiring dose reductions due to adverse events (versus 17% and 3% in the placebo arm, respectively) [52]. Dose reductions were most commonly needed for anemia, fatigue, nausea, and neutropenia, the majority of which resolved following dose modification and supportive care. Treatment continued for at least 2 years in 57.3% of patients in the olaparib arm (47% completed 2 years and 10% received treatment beyond 2 years). Of these patients, 64% were receiving the recommended starting dose of 300 mg olaparib, twice-daily. At the 2-year time point, when most patients without disease recurrence completed 2 years of maintenance treatment, there was no change in the rate of disease progression [8].

In the PRIMA trial, all patients were initially given a fixed dose of 300 mg once-daily; however, the protocol was amended to start patients on either niraparib 200 or 300 mg once-daily based on weight and platelet count. A post-hoc analysis of the ENGOT-OV16/NOVA trial identified weight <77 kg and platelet count <150,000/μL to be risk factors for grade ≥3 thrombocytopenia, while a descriptive analysis of PFS after 4 months found that reduction of niraparib doses to 200 mg and 100 mg did not impact PFS [60]. The fixed dosing strategy used in the first 315 of 484 patients who received niraparib may have contributed to the high rate of adverse events leading to dose reductions (71%) and dose interruptions (80%) [9]. The individualized dosing strategy was found to reduce rate of adverse events. In addition, a post-hoc analysis of the PRIMA study found the HR for progression or death in patients receiving niraparib versus placebo was comparable in the subgroups of patients treated with fixed dosing and individualized dosing strategies [61]. A retrospective, observational study in 153 American patients with recurrent EOC who received niraparib 200 mg as a starting dose found that in the real-world setting, this lower starting dose resulted in a lower frequency of common adverse events that were observed in the NOVA and PRIMA trials [62]. In addition, 26% of patients were subsequently able to increase their dose to 300 mg/day. This study did not assess the impact of this dosing strategy on progression outcomes.

### 5.3. Interpretation and Canadian Perspective

The panel recommends following the suggested dosing described in the product monographs for each PARP inhibitor, including the personalized dosing strategy for niraparib, and any dose reductions suggested in patients with hepatic insufficiency (niraparib), renal insufficiency (olaparib), or co-administration with CYP3A inhibitors (olaparib) [58,59]. Dose interruptions or reductions may be needed for toxicity management and should follow the dose adjustments recommended in the product monographs. Initial dosing may be resumed following resolution of toxicity. In general, dose interruptions should be considered in patients with persistent grade ≥3 adverse events up to 28 days or until the toxicity is resolved [63]. Dose modifications may also be considered in patients with grade 1 or 2 adverse events if symptoms do not improve despite management strategies, or if they are intolerable to the patient.

Patients with no evidence of disease may feel anxious about completing PARP inhibitor treatment at the indicated 24-month (olaparib) or 36-month (niraparib) time-point. It is important to discuss the planned duration of therapy with patients at the time of PARP inhibitor initiation. Patients should be informed of the data to support the duration of maintenance therapy. There are currently no clinical trials evaluating whether PARP inhibitor maintenance beyond the currently indicated durations would improve survival outcomes in patients with EOC or increase risks of toxicities, including MDS/AML. The rates of MDS/AML reported in clinical trials of olaparib and niraparib are higher in patients receiving PARP inhibitors for relapsed disease versus in the first-line maintenance setting [8,9,55,64]. The increased risk of MDS/AML in patients with recurrent EOC may be due to the greater prior exposure to chemotherapy, in addition to the PARP inhibitor treatment or presence of BRCA1/2 mutations.

Given the frequent incidence of hematological adverse events during PARP inhibitor treatment and the risk of MDS/AML, it is imperative to have a follow-up schedule that will allow for the monitoring of these and other adverse events. The panel recommends monthly visits for up to 6–12 months as needed based on the patient's symptoms or clinical characteristics, followed by every 2–4 months thereafter. Complete blood count assessment should be performed monthly up to 24 months (weekly during the first month of niraparib treatment), then every 3–4 months at routine visits. Regular blood pressure monitoring at home is suggested for patients receiving niraparib as grade ≥3 hypertensive events have been reported in 6–8% of patients [9,27].

If a toxicity arises during PARP inhibitor maintenance that cannot be appropriately managed, it is reasonable to consider switching to a different PARP inhibitor, where accessible, as this may allow for the continuation of therapy. Although unstudied, this practice has been deemed acceptable by the Canadian Agency for Drugs and Technologies in Health [65].

## 6. Conclusions

A decision-making algorithm for maintenance therapy in patients with advanced, high-grade EOC after response to first-line platinum-chemotherapy, based upon current evidence and Canadian access to testing and therapies, is depicted in Figure 1. It highlights the need for timely access to genomic testing including tumour and germline BRCA1/2 mutation assessment and HRD assays to guide PARP inhibitor selection. PARP inhibitor maintenance therapy demonstrates substantial prolongation of PFS in patients with BRCA1/2 mutations, with approximately 50% of patients treated with olaparib in the SOLO-1 trial being free of recurrence at a median follow-up of 5 years. PARP inhibitors are thus the preferred first-line maintenance treatment for BRCA1/2-mutated EOC. Confirmation of HRD with current genomic scar assays, although imperfect at differentiating all patients who may benefit from PARP inhibitor maintenance, can help identify patients with advanced EOC who are most likely to benefit from this treatment and provide data to make informed decisions regarding maintenance therapy. However, the recommendation for PARP inhibitor maintenance therapy may change as survival data mature. Standardization and optimization of HRD tests, a better understanding of mechanisms of resistance to PARP inhibitor therapy, and improved therapies for patients with HRP will increase the utility of HRD testing for guiding treatment decisions. New therapeutic strategies for first-line advanced EOC continue to be investigated, including use of PARP inhibitors as induction treatment, PARP inhibitor combinations with immune checkpoint inhibitors, and novel agents targeting other DNA damage response pathways. These may further complicate treatment algorithms and necessitate updated tools to aid Canadian oncologists with decision-making for their patients with advanced EOC.

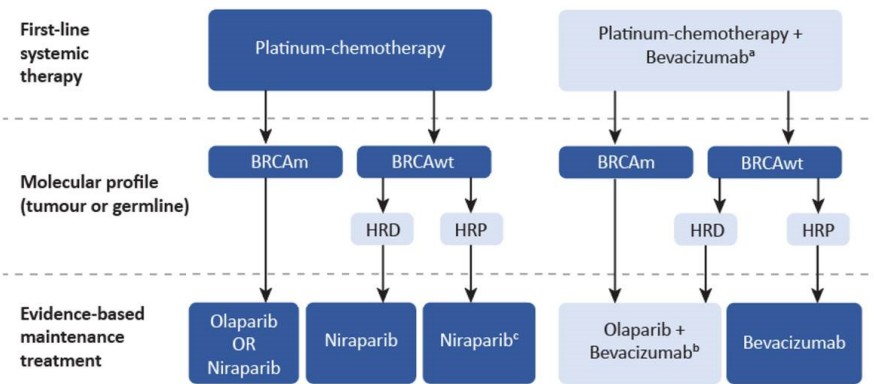

**Figure 1.** Maintenance treatment options for advanced stage, high-grade EOC following response to first-line platinum-based chemotherapy. EOC, epithelial tubo-ovarian cancer; HRD, homologous recombination repair deficiency; HRP, homologous recombination repair proficiency; m, mutated; wt, wild-type. Light blue indicates testing/treatment with variable access in Canada as of March 2022. [a] Funded in most provinces in the first-line setting for patients with high-risk disease defined as stage III sub-optimally debulked, stage III unresectable, or stage IV disease. [b] Not approved by Health Canada. [c] Weigh risks/benefits of starting niraparib on individual basis considering patient and disease factors.

**Author Contributions:** Conceptualization, A.V.T. and A.M.O.; writing—original draft preparation, S.D. and A.V.T.; writing—review and editing, A.V.T., A.D.A., M.Q.B., P.G., L.T.G., D.P., S.S. and A.M.O. All authors have read and agreed to the published version of the manuscript.



**Funding:** Funding was provided by AstraZeneca Canada to support medical writing assistance and administrative coordination of this manuscript. The funders did not contribute to the content or writing of the manuscript.

**Acknowledgments:** The authors acknowledge medical writing support, provided by Sarah Doucette of IMPACT Medicom, which was funded by AstraZeneca Canada.

**Conflicts of Interest:** Anna V. Tinker reports institutional grants from AstraZeneca and personal fees from AstraZeneca, GlaxoSmithKline, and Eisai. Alon D. Altman reports speaker fees from Roche, Sanofi, AstraZeneca, and Merck, as well as served on advisory boards for AstraZeneca Canada and GlaxoSmithKline. He has also received research grants as a principal investigator for clinical trials from AstraZeneca, Pfizer, Clovis, Array Biopharmaceutical, INNOVATE, and the National Cancer Institute of Canada. Prafull Ghatage has served on advisory boards for AstraZeneca Canada, GlaxoSmithKline, and Eisai. Diane Provencher has served on advisory boards for Clovis Oncology, AstraZeneca, GlaxoSmithKline, Roche-Genentech, and Tesaro. Shannon Salvador has served on advisory boards for GlaxoSmithKline, AZ, and Merck. She holds institutional grants for research from GlaxoSmithKline and AstraZeneca. Sarah Doucette has received funding from AstraZeneca Canada for medical writing support on this manuscript. Amit M. Oza reports institutional research grants from AstraZeneca; has served on an advisory board (uncompensated) for GlaxoSmithKline; has served on advisory boards and steering committees (uncompensated) for Clovis Oncology and AstraZeneca; and has served as a principal investigator on investigator-initiated trials for Clovis Oncology, AstraZeneca, and GlaxoSmithKline. All other authors declare no competing interests.

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
