# Peer review of "A Pan-Canadian Consensus Statement on First-Line PARP Inhibitor Maintenance for Advanced, High-Grade Serous and Endometrioid Tubal, Ovarian, and Primary Peritoneal Cancers"

_curroncol, doi:10.3390/curroncol29060348_

Round 1

Reviewer 1 Report

Thank you for your review of current data to support the use of PARP inhibitors in front line therapy for Ovarian cancer. This paper gives a Canadian lens on this and is relevant to the readership of this journal. There are reviews published but this fills a gap in this particular context. Access to this therapy and knowledge translation to all areas where ovarian cancer patients are treated is very important. As stated, it is clear and the conclusions are supported by most of the evidence. Several points were noted: Line 122 states that the BRCA mutation rate was up to 23 % are there any contemporary Canadian figures, this seems high? Line 426 the dose of Olaparib was stated at 300mg daily, this should be twice daily. In Table 2 you list all the trials including the relapsed setting, this does not add to the paper without a discussion of when to use second line PARP, or is there any value of waiting to recurrence before using a PARP inhibitor. If not this section of the table should be considered for removal.

Author Response

Thank for providing these thoughtful comments.

To address the first comment regarding rate of germline BRCA mutation in high-grade serous ovarian cancer, we have changed the 23% percent originally stated on line 122 to include a range of 15-25% (now appearing on line 124). Upon reviewing the literature for contemporary/Canadian figures, the BRCA mutation rate appeared to vary within this range by geographical location and population studied. A study based in British Columbia showed a rate of 21% for serous ovarian cancer. Another study looking at French Canadian patients with ovarian cancer from Quebec reported a rate of 19.2% for high-grade serous ovarian cancer. A recent meta-analysis looking at 28 studies published between 2015-2020, based on populations in North America, Europe, and Asia, found the rate of BRCA mutation in high-grade serous ovarian cancer to be 22%. These Canadian studies and meta-analyses have been included in the reference list.

The second comment identified an error on line 426 where olaparib was stated as being given at a dose of 300 mg daily. This has been changed to say “twice-daily”.

For the final comment, the authors agreed with the suggestion of removing the trials related to the relapsed setting, which can be seen in the tracked changes on page 6. As this paper focusses on PARP inhibitor maintenance therapy in the first-line setting, a discussion of when to use PARP inhibitors in the second-line setting to accompany this portion of the table is out of scope.  

Reviewer 2 Report

Suggestion 1: The high-grade EOC may be specifically defined as serous or endometrioid carcinoma given that the data in this article are mainly based on SOLO1 and PRIMA studies. Only patients with high-grade serous or endometrioid carcinoma were included in these two studies.

Suggestion 2: Primary serous peritoneal carcinoma is suggested to be included in the title of the article.

Author Response

Thank you for providing these thoughtful comments.

The first suggestion acknowledged that high-grade EOC should be more specifically defined as serous of endometrioid carcinoma, as these are the subtypes that were evaluated in the SOLO1 and PRIMA trials. This has been addressed on line 59 with the following sentence: “"High-grade EOCs, which include serous or endometrioid carcinomas, are particularly susceptible to the cytotoxic effects of platinum-based chemotherapy [5, 6]."

The second suggestion also involved more specifically stating the population of ovarian cancer that will be discussed in the paper by including primary serous peritoneal carcinoma in the title. Based on this suggestion, the title has been changed to: “A pan-Canadian consensus statement on first-line PARP inhibitor maintenance for advanced, high-grade serous and endometrioid tubal, ovarian, and primary peritoneal cancers”

Reviewer 3 Report

Anna V. Tinker et al., reported that the PARP inhibitors (Olaparib and niraparib) have been established as a standard chemotherapy in EOC in Canada. The authors clearly discuss about the clinical evidence for first-line PARP inhibitors especially in High grade EOC. The background and explanation for the clinical evidences every this is good and useful for further development of clinical studies.

I have two generalized comments/questions  

1.    As we know that and stated in abstract, 80% of patients will experience a recurrence after receiving platinum based chemotherapy, what is the percentage of recurrence after receiving the PARP inhibitors (Olaparib and niraparib) ?

2.    I feel authors didn't adequately explain the effect of the PARP inhibitors in metastases ?

Author Response

Thank you for providing these thoughtful comments.

As mentioned in the first comment, the paper acknowledges that 80% of patients with advanced, high-grade ovarian cancer will experience a recurrence after platinum-based therapy, and this is used as a rationale for the use of PARP inhibitors to extend progression-free survival and reduce the rate of recurrence. We agree that the rate of recurrence following first-line PARP inhibitor maintenance should therefore be more clearly stated. There is long-term data for the rate of recurrence with olaparib maintenance in patients with BRCA1/2 mutations from SOLO-1. At a median follow-up of 5 years, approximately 50% of patients in the olaparib arm had not experienced disease progression compared with 20% who had not experienced disease progression in the placebo arm. This data is cited by risk subgroup on lines 273-275; however, to increase clarity, the following sentence has been added to the summary and conclusions section: “PARP inhibitor maintenance therapy demonstrates substantial prolongation of PFS in patients with BRCA1/2 mutations, with approximately 50% of patients treated with olaparib in the SOLO-1 trial being free of recurrence at a median follow-up of 5 years.” (Lines 500-503)

There are caveats to drawing conclusions on recurrence rates based on the short-term data published in the PRIMA study. This is accentuated by the recent notice letter issued to Health Care Providers by the manufacturer of niraparib stating that the updated survival data from the NOVA study signal a potential detriment of niraparib maintenance on overall survival for patients without BRCA mutations. At the current median follow-up in the PRIMA study of 13.8 months, the estimated rate of progression-free survival at 18 months was 59% and 35% for patients with HRD in the niraparib and placebo arms, respectively (42% and 28% for the overall population). This early data suggests that recurrence rates may be reduced by the use of PARP inhibitors, but longer-term data on PFS and OS will be needed to confirm that the effect is durable and translates to a survival advantage in all subgroups (e.g BRCA, HRD, non-BRCA, non-HRD etc.). Because of this uncertainty, the following sentence was added to the summary and conclusions section regarding the benefit of PARP inhibitor maintenance in the HRD population: “However, the recommendation for PARP inhibitor maintenance therapy may change as survival data mature.” (Lines 508-509)

The second suggestion to explain the effect of PARP inhibitors more thoroughly in metastatic disease was challenging to address as this specific data was not reported in the SOLO1 or PRIMA studies. We have however, included a sub-analysis of the SOLO1 trial by disease risk (lines 273-275), where among the 55% of patients with high-risk disease, 5-year PFS rates for olaparib and placebo were 42.1% and 17.3%, respectively. Among the high-risk patients, 31% of patients with stage IV disease, as well as patients with stage III disease after neoadjuvant chemotherapy and interval cytoreduction or with gross residual disease following initial cytoreductive surgery were included. The PRIMA trial does include a sub-analysis of PFS by disease stage, which reports a hazard ratio of 0.79 (95% CI 0.55-1.12) for niraparib versus placebo in patients with stage IV disease. However, this analysis is based on the overall population, rather than the HRD and BRACm populations where PARP inhibitors are expected to be the most beneficial. For this reason, this data was not included in the paper.